# Inhibition of Phytosterol Biosynthesis by Azasterols

**DOI:** 10.3390/molecules25051111

**Published:** 2020-03-02

**Authors:** Sylvain Darnet, Laetitia B B Martin, Pierre Mercier, Franz Bracher, Philippe Geoffroy, Hubert Schaller

**Affiliations:** 1CVACBA, Instituto de Ciências Biológicas, Universidade Federal do Pará, Belém 66075-750, Brazil; sylvain@ufpa.br; 2Plant Isoprenoid Biology (PIB) team, Institut de Biologie Moléculaire des Plantes du CNRS, Université de Strasbourg, 67084 Strasbourg, France; Laetitia.Martin@jic.ac.uk (L.B.B.M.); pierre.mercier@ibmp-cnrs.unistra.fr (P.M.); 3Department of Pharmacy, Center for Drug Research, Ludwig-Maximilians University Munich, 81377 Munich, Germany; franz.bracher@cup.uni-muenchen.de; 4Institut de Chimie, Université de Strasbourg, 67008 Strasbourg, France; p.geoffroy@unistra.fr

**Keywords:** cholesterol, sitosterol, fungicide, selective herbicide, SBI, steroidogenesis inhibitor

## Abstract

Inhibitors of enzymes in essential cellular pathways are potent probes to decipher intricate physiological functions of biomolecules. The analysis of *Arabidopsis thaliana* sterol profiles upon treatment with a series of azasterols reveals a specific in vivo inhibition of SMT2, a plant sterol-C-methyltransferase acting as a branch point between the campesterol and sitosterol biosynthetic segments in the pathway. Side chain azasteroids that modify sitosterol homeostasis help to refine its particular function in plant development.

## 1. Introduction

Sterol biosynthesis, is an essential cellular metabolic pathway well-described in many eukaryotic organisms and also in some bacteria bearing minimal sterol pathways [1,2,3]. Plants transform the committed isoprenoid-derived 2,3-oxidosqualene precursor into the tetracyclic steroidal triterpene cycloartenol, whereas fungi and mammals produce lanosterol from the same precursor (Figure 1A). These bioconversions are catalyzed by two distinct oxidosqualene cyclases (OSCs), cycloartenol synthase (EC 5.4.99.8) and lanosterol synthase (EC 5.4.99.7), respectively [4,5,6]. The presence of lanosterol synthases and lanosterol in plants has been reported; the major function of lanosterol as a biosynthetic intermediate or functional metabolite in species like *Arabidopsis thaliana* [7] or *Euphorbia lathyris* [8] is, however, not yet very well understood. Cycloartenol bears a 9β,19-cyclopropane moiety on ring B of the tetracyclic structure, which is metabolized by the cycloeucalenol isomerase (CPI, EC 5.5.1.9; Figure 1A) to form obtusifoliol during the conversion of sterol precursors into the end-products -^5^-sterols [9]. Obtusifoliol is a 14α-methyl-^8^-sterol biosynthetic intermediate that shares structural features with lanosterol, another type of 14α-methyl-^8^-sterol. These compounds are substrates of cytochrome-dependent P450 monooxygenases, obtusifoliol-14-demethylase (EC 1.14.13.70), and lanosterol-14-demethylase (EC 1.14.13.70) in plants, mammals, and fungi, respectively [10]. The biological significance of the mandatory and sophisticated biogenetic detour from cycloartenol to obtusifoliol, in the case of plants, has been linked to specific aspects of pollen lipid biology [11]. Plants exhibit further specific aspects of sterol biology as compared with other eukaryotes. The enzymatic transformation of cycloartenol to -^5^-sterols (cholesterol, campesterol, sitosterol, and stigmasterol) implies the oxidative removal of two methyl groups at C-4 of the tetracyclic sterol nucleus (Figure 1A) [12]. These two demethylation reactions occur on lanosterol in mammals and fungi in a sequential manner [13] but are not consecutive in the plant pathway. In contrast, plants display successive 4,4-dimethyl sterols, 4-methylsterols, and 4-desmethylsterols biosynthetic segments. An exhaustive state-of-the-art of the biosynthetic and physiological implications of 4-methylsterols was recently published [2]. Furthermore, the addition of two exocyclic carbon atoms in the side chain of sterol substrates to generate 24-methyl(ene)sterols and 24-ethyl(idene)sterols (such as 24-methylcholesterol and sitosterol, respectively, Figure 1B) is one of the most studied types of enzymatic reactions in sterol biochemistry [14] and is also a significant feature of land plant sterol biosynthesis [15]. Two distinct S-adenosyl-L-Met-sterol-C24-methyltransferases (EC2.1.1.41), i.e., (sterol-C24-methyltransferases, SMTs), are responsible for two non-consecutive methyl transfers in the conversion of cycloartenol to sitosterol. SMT1 catalyzes the methylation of cycloartenol at C-24 to yield 24-methylene cycloartenol, and SMT2 catalyzes the methylation of 24-methylenelophenol at C-24^1^ to produce 24-ethylidenelophenol. Contrastingly, fungal sterols have a single exocyclic carbon atom in their side chains, and mammalian sterols have none [16]. The biological significance of distinct SMTs in plants was addressed by the characterization of loss-of-function mutations; significant morphogenetic inhibitions were observed in the case of impaired *SMT2* gene expression [17].

The sterol biosynthesis pathway contains multiple enzymatic targets for inhibitory molecules grouped into main categories, such as piperazine, morpholine, pyridine, pyrimidine, and azole derivatives [22]. Some of these chemicals, such as azole and morpholine fungicides, are widely used in medicine or agriculture based on their potent inhibitory action of the enzymes lanosterol-14-demethylase, as well as sterol-8-isomerase (SI, EC5.3.3.5) and sterol-14-reductase (14R, EC1.3.1.70). Numerous studies on the activity and mode of action of these compounds on sterol biosynthesis enzymes of mammalian [23], fungal [23], or parasitic origin have been performed and are continuously going on [24]. The interest in finding new compounds of synthetic or natural origin and modifying their structure to improve their efficiency remains unbroken, although certain enzymes like the fungal sterol-22-desaturase (EC 1. 14. 19. 41) did not efficiently comply with the criteria of interesting new drug targets [25,26,27,28]. Recently, the characterization of a natural steroidal inhibitor of a sterol-4α-carboxylate-3-dehydrogenase, an enzyme of the sterol-C4-demethylation complex from yeast (C4DMC) clearly indicated that many target enzymes had been overlooked so far in chemical and pharmaceutical screenings for new bioactive ligands [29,30].

Here, the focus is on several enzymes of the sterol pathway, which all imply carbocationic high energy intermediates during their catalytic process [31]. In fact, OSCs, CPI, and SMTs are inhibited by rationally designed stable analogs of these carbocationic intermediates [32]. Comprehensive enzymological studies depicting the features of carbocationic mimicks have previously highlighted the powerful effect of these inhibitors to control in vivo the sterol profiles of plant cells and organisms and to implement lead molecules as novel classes of rationally designed inhibitors of considerable value for agronomical applications [33]. Despite active in vitro effects, most of the substrate analogs (mechanism-based inactivators, high energy carbocationic intermediates) were shown to hit several enzymatic targets in the sterol pathway, for example, 25-aza-cycloartanol was shown to act on both SMT1 and SMT2 enzymes in plant cells [34].

In this article, the effect on *Arabidopsis thaliana* seedlings of a nitrogen-containing hydrocarbon and a series of side chain azasteroids (also called aminosterols) is reported. The specific in vivo inhibition of the land plant enzyme SMT2 by some azasterols is confirming these latter as valuable probes to study the function of distinct phytosterol profiles at multiscale levels.

## 2. Results

### 2.1. Multiple Target Sites of LDAO

*Arabidopsis thaliana* grown on a synthetic medium containing LDAO exhibited a dramatic growth arrest at the early stages of seed germination at a concentration of 20 mg·L^−1^ (85 μM). The striking feature of the growth inhibition is the severe bleaching of seedlings at the cotyledonous stage at the highest concentration assayed (Figure 2A,B). The chemical profiles of whole seedlings established by GC-FID and GC-MS showed that 2,3-oxidosqualene and also 9β,19-cyclopropylsterols were predominant in treated material, whereas these compounds were detected at very low levels in untreated control seedlings, in which a typical plant sterol profile made of 24-methyl and 24-ethyl -sterols (campesterol, isofucosterol, sitosterol, and stigmasterol) was established (Table 1).

LDAO has been reported as one of the most potent inhibitors of 2,3-oxidosqualene cyclization into lanosterol, cycloartenol, or β-amyrin [35]. In the present study, the accumulation of 9β,19-cyclopropylsterols in seedlings up to 80% of the total sterols clearly, and for the first time, shows that CPI is a target site of LDAO in plants in addition to the OSC cycloartenol synthase. The molecular features of LDAO, which carries a positively charged nitrogen atom located at the extremity of a saturated hydrocarbon chain (Figure 2C) is readily amenable to stable conformations acting as postulated carbocationic high energy intermediates at play in the cyclization and isomerization reactions performed by CAS and CPI (Figure 1A), respectively. When fed to plant cells or germinating seeds, LDAO acts at the same time as an analog of the protonated 2,3-oxidosqualene to inhibit CAS [35,36] and as an analog of the C-9 carbocationic intermediate of the enzymatic conversion of cycloeucalenol into obtusifoliol [37] (Figure 2C). The dual target of LDAO in plant sterol biosynthesis confers powerful biocidal properties to the compound but, at the same time, prevents its use as a specific molecular tool to control the pathway at a given step. This situation is reminiscent of the mode of action of morpholines, for instance, amorolfine [38] and tridemorph, which inhibit more than one enzyme in ergosterol biosynthesis. It is also reminiscent of other types of growth inhibitors such as triazole-type fungicidal compounds that inhibit cytochrome P450 monooxygenases (lanosterol-14-demethylase in fungi), which have been shown to target plant P450 enzymes implied in phytosterol, brassinosteroid, or gibberellin pathways [39].

### 2.2. Selective Inhibition of SMTs

A search for specific inhibitors of plant sterol-C-methyltransferases was conducted using the simple assay with *Arabidopsis thaliana* seedlings, as described above. Sterol analogs S1 to S17 (Figure 3), already known as inhibitors of fungal sterol-C24-methyltransferase by Renard et al. [40], were fed to germinating seeds at a concentration of 2 mg·L^−1^ (about 5 μM). Growth inhibition was observed after two to four weeks of growth according to the aspects of small rosettes and roots, which were defined as four categories of biological effect and, consequently four categories of compounds A, B, C, and D (Figure 4). Firstly, the general aspect of treated seedlings highlighted the strong inhibition of root development as a visible initial growth defect. Total root length was about 70% of the control length in category A of sterol analogs comprising compounds S5 to S10; about 15% of the control root length in category B comprising compounds S15, S16, and S17; about 90% in category C comprising compounds S11 and S12; and over 90% in category D comprising compounds S13 and S14, for which the most severe conditions were observed (Figure 4). Compounds S1 to S4 had no visible effects on growth (not shown).

Second, whereas A, B, and C cultures looked at from a top view did not show any dramatically hampered rosette development, seedlings belonging to the D category displayed leaf tissue damage, in addition to the negative action on roots, one month after the onset of germination.

Third, sterol profiles were determined at 15 days after germination in the case of all sterol analogs S1 to S17 considered in the present experiment. Categories of chemical profiles were easily defined according to the proportion of 24-desmethylsterols (sterols bearing a C_8_ side chain), 24-methylenesterols and 24-methylsterols (sterols with a C_9_ side chain), and 24-ethylidenesterols and 24-ethylsterols (sterols with a C_10_ side chain), indicative of the inhibition of SMT1 or SMT2 (Figure 1B and Table 2). Typical GC-FID traces produced for each type of sterol profile Control A, B, C, and D are shown in Figure 5. Group A seedlings contained about 25% to 30% of sterols with a C_9_ side chain (mostly 24-methylcholesterol) and about 50% to 60% of sterols with a C_10_ side chain (mostly sitosterol). Seedlings treated with compounds from the category B (S15, S16, and S17) displayed a remarkable inversion of the proportion of 24-methylcholesterol to that of sitosterol without much effect on the amount of cycloartenol and cholesterol, pointing at a potent and specific inhibitory effect on SMT2 in vivo (Table 2 and Figure 5B). Clearly, the sterol profiles of such treated *Arabidopsis* seedlings are copies of the sterol phenotypes of loss-of-function alleles of *SMT2* genes [15,17]. Seedlings from groups C and D exhibited a rather moderate increase in the amount of 24-methylcholesterol, a decrease in sitosterol, and an increase in cycloartenol and cholesterol, a chemical profile indicative of a possible dual inhibitory effect on both SMT1 and SMT2 (Table 2 and Figure 5C). Furthermore, seedlings from the category D (S13 and S14) had a very strong enrichment in cycloartenol and cholesterol, the major C_8_ side chain sterols (Table 2 and Figure 5D). The predominance of cycloartenol and cholesterol in such sterol profiles, which is reminiscent of the sterol profile of a loss-of-function allele of SMT1 [41], provides conclusive evidence of a strong inhibition of SMT1 by group D compounds. In such *smt1-1 Arabidopsis thaliana* mutant, the proportion of C_9_-side chain sterols is slightly higher in the mutant (31%) versus the wild-type (25%) [41]. Consequently, the increase in 24-methyl(ene) sterols in groups C and D could also be caused from yet unspecified effects downstream to SMT1.

The in vivo efficient blockage of SMT1, which is acting upstream to SMT2 in the plant sterol pathway, is possibly masking a potential in vivo effect on SMT2, an assumption that would need a thorough research effort focusing on exhaustive investigations of purified SMT1 and SMT2. These inhibitors produce the deleterious effect shown in Figure 4D because of the drastic reduction in the proportion of 24-alkyl-^5^-sterols (C_9_ and C_10_ side chain sterols), which are necessary to sustain cellular sterol homeostasis and plant growth [3,15,42]. It is worth noting that cholesterol, although present in all plant species at low concentrations, is abundant in species from the Solanaceae that contain a high amount of cholesterol-derived steroidal glycoalkaloids [43].

## 3. Discussion

The sterol profiles of *Arabidopsis thaliana* seedlings grown in the presence of stable analogs of carbocationic high energy intermediates implied in several enzymatic conversions of sterol substrates demonstrates multiple enzymatic target sites for these compounds. These enzymes are possibly phylogenetically unrelated similar to the case for cycloartenol synthase (2,3-oxidosqualene cyclase) and cyclopropyl isomerase, or closely related like cycloartenol-C24-methyltransferase and 24-methylenelophenol-C24^1^-methyltransferase, SMT1 and SMT2, respectively. The assays performed in this study have revealed LDAO as an inhibitor of CPI in addition to its capacity to block the cyclization of 2,3-oxidosqualene into cycloartenol. This qualifies LDAO as a multisite inhibitor similar to other compounds extensively characterized [29,33]. However, it is crucial to identify selective inhibitors of the sterol pathway in order to link a given sterol profile to a biological process in organs or cell types. For this reason, we further looked at sterol analogs interfering with the addition of exocyclic carbon atoms on the sterol side chains. The phylogenetic classification of SMTs clearly distinguishes SMT1 from SMT2 types of enzymes (Figure 6). SMT1s cluster with the orthologous ERG6 and other fungal SMTs, enabling the formation of the C_9_ sterol side chain of ergosterol and other 24-methylsterols, whereas SMT2s characterized as typical land plant catalysts are responsible for the formation of C_10_ side chain sterols such as β-sitosterol [15]. The first output of the seedling growth assays, reported here, is the selective inhibition of SMT2 by the azasteroids S15, S16, and S17. These three azasterols possess a tetracyclic sterol nucleus with a double bond at C-7(8) and a 3β-hydroxy group such as 24-methylenelophenol, the natural substrate of SMT2 [16,44,45] and a C-17 side chain with an overall size similar to the side chains of the physiological sterols containing a protonable aliphatic amino group. Amino groups (or modifications such as diamines and pyridinium ions), which enable the compounds to mimick carbenium ions in vivo, are also present in the other steroids (Figure 3) as well, obviously slight structural variations can lead to significant changes in biological activities. The activity profiles of the investigated azasteroids are similar to those reported before for fungal SMT [40].

Further comparative structure-function relationship studies are required in order to characterize the intrinsic properties of the inhibitors of SMT1 and SMT2, especially to understand the electrostatic interactions at play that bind inhibitors to the enzyme. Nevertheless, these side chain azasteroids are actually valuable molecular functional probes for controlling the production of sitosterol and characterizing its major effect on root development. Overall, it is shown here that azasterols rank as lead compounds exhibiting interesting specificity towards steroidogenic enzymes. Furthermore, it is quite striking to observe that functional azasterols occur naturally in a few organisms, such as the case of the telluric fungus *Geotrichum flavo-brunneum* which produces a natural 15-azasterol (15-aza-24-methylene-D-homocholestadiene-3β-ol (Figure 7) [46]. This compound is a suitable inhibitor of the sterol-14-reductase in yeast and other fungi based on structural and charge analogies between its protonated form at cellular pH and the C-14 carbocationic high energy intermediate of the reaction that converts a -^8,14^-sterol into a -^8^-sterol [47]. The natural occurrence of such inhibitors is possibly part of the chemical arsenal of ecological interactions.

Finally, the inhibition of sterol-C24-methyltransferases is appealing outside of the plant metabolic biology field because humans (mammals) do not possess such SMT. In fact, chemicals that can be exploited in chemotherapy, especially to fight pathogenic fungi such as *Cryptococcus*, but also protozoa such as *Trypanosoma* or *Leishmania*, have been characterized [25,48,49,50].

## 4. Material and Methods

### 4.1. Chemicals

LDAO (C_14_H_31_NO, lauryldimethylamine oxide, and N,N-dimethyldodecylamine N-oxide) was purchased from Serva. Chemical synthesis of azasterols S1 to S17 considered in this work has been described in detail in Renard et al. [40]. All steroidal test compounds share a common structural feature, which is a steroidal tetracyclic moiety bearing a double bond at C-7(8). Structural specificities of S1 to S17 are given by side chains of various lengths at C-17, the position of a heteroatom (N) in the side chain (at position C-23, C-24, C-25, C-26, or C-27), or a pyridinium ring at C-23. Position C-3 is substituted by 3β-hydroxy, 3β-acetate, or 3β -amino-methyl groups. Nomenclature as follows and as in Renard et al. [40]:**S1**, C_34_H_61_NO, CAS 1202777-29-4: Pregn-7-en-3-ol, 21-(dodecylamino)-20-methyl, (3β, 5α, 20S)-;**S2,** C_36_H_63_NO_2_, CAS 1202777-41-0: Pregn-7-en-3-ol, 21-(dodecylamino)-20-methyl, 3-acetate, (3β, 5α, 20S)-;**S3**, C_27_H_48_N_2_, Pregn-7-en-3-aminomethyl-21-(butylamino)-20-methyl-, (3β, 5α, 20S)-;**S4**, C_26_H_46_N_2_O, CAS 1202777-19-2: Pregn-7-en-3-ol, 21-[[2-(dimethylamino) ethyl] amino]-20-methyl, (3β, 5α, 20S)-;**S5**, C_29_H_42_NOI, Pyridinium, 4-[(3β, 5α, 22Z)-3-hydroxy-24-norchola-7,22-dien-23-yl]-1-methyl-iodide (1:1);**S6**, C_29_H_42_NOI, Pyridinium, 3-[(3β, 5α, 22E)-3-hydroxy-24-norchola-7,22-dien-23-yl]-1-methyl iodide (1:1);**S7**, C_26_H_43_NO, 26, 27-dinorcholesta-7,22-dien-3-ol, 25-(dimethylamino), (3β, 22Z)-;**S8**, C_29_H_49_NO_2_, CAS 1202777-10-3: Pregn-7-en-3-ol, 21-(butylmethylamino)-20-methyl-3-acetate, (3β, 5α, 20S)-;**S9**, C_28_H_47_NO_2_, CAS 1202777-37-4: Pregn-7-en-3-ol, 21-(butylamino)-20-methyl-3-acetate, (3β, 5α, 20S)-;**S10**, C_28_H_47_NO_2_, CAS 1202777-39-6**:** Pregn-7-en-3-ol, 20-methyl-21-[(2-methylpropylamino]-3acetate, (3β, 5α, 20S)-;**S11**, C_29_H_42_INO, CAS 1202777-30-7: Pyridinium, 2-[(3β, 5α, 22E)-3-hydroxy-24-norchola-7,22-dien-23-yl]-1-methyl-iodide (1:1);**S12**, C_29_H_42_NOI, CAS 1001324-61-3: Pyridinium, 4-[(3β, 5α, 22E)-3-hydroxy-24-norchola-7,22-dien-23-yl]-1-methyl-iodide (1:1);**S13**, C_26_H_45_NO, CAS 1202777-24-9: Pregn-7-en-3-ol, 20-methyl-21-(propylamino) (3β, 5α, 20S)-;**S14**, C_26_H_45_NO, CAS 1202777-25-0: Pregn-7-en-3-ol, 21-(butylamino)-20-methyl-(3β, 5α, 20S)-;**S15**, C_25_H_41_NO, chola-7,22-dienol-24-(dimethylamino) (3β, 5α, 22Z);**S16**, C_24_H_41_NO, CAS 1202777-23-8: Pregn-7-en-3-ol, 21-(ethylamino)-20-methyl-(3β, 5α, 20S)-;**S17**, C_27_H_47_NO, CAS 1202777-28-3: Pregn-7-en-3-ol, 21-[(1,2-dimethylpropyl)amino]-20-methyl-(3β, 5α, 20S)-.

### 4.2. Plant Treatment with Chemicals

*Arabidopsis thaliana* (L) Heynh ecotype Col-0 seeds were germinated on synthetic medium (Murashige and Skoog medium M0221 purchased from Duchefa, standard Petri plates) containing a given concentration of LDAO or a concentration of 2 mg·L^−1^ of each of the azasteroids S1 to S17. Azasteroids were stored in ethanol at a concentration of 2 mg·mL^−1^. Control treatments consisted of 1 mL·L^−1^ of ethanol. The effect of the compounds on seedling growth was assessed at 15 days and 30 days after germination. Experiments were done twice independently and in a triplicate assay for each compound. One of these triplicates in each biological assay was analyzed to determine its sterol profile.

### 4.3. Sterol Analysis

Freeze-dried seedling samples (100 mg) were saponified for one hour in a 6% potassium hydroxide methanolic solution at 80 ℃. The unsaponifiable fraction was extracted with three portions of hexane. The dried extract was acetylated with a mixture of pyridine/acetic anhydride in toluene. Steryl acetate detection, analysis, and quantification were performed as described [51,52]. In brief, the quantification was performed with a 3400CX FID-based gas chromatograph (Varian). The DB-5 column for the GC analysis has the following features: 30 m wall-coated open tubular, 250 µm film thickness, and 0.32 mm internal diameter. The H_2_ flow rate was fixed at 2 mL.min^−1^ and the injector and detector temperatures were 250 ℃ and 300 ℃, respectively. The GC-MS identification of the compounds was performed using a 5973N instrument (Agilent) equipped with a primary HP5-MS column and coupled with a 6853 mass analyzer (Agilent). The temperature program of ovens (GC-FID and GC-MS) has a two-step phase: (i) a steep ramp at 30 ℃/min from 60 ℃ to 220 ℃ and (ii) a 2 ℃/min^−1^ increase from 220 ℃ to 300 ℃. Sterols were identified by coincidental retention time and EI-MS spectra at 70 eV [53].

## Figures and Tables

**Figure 1 molecules-25-01111-f001:**
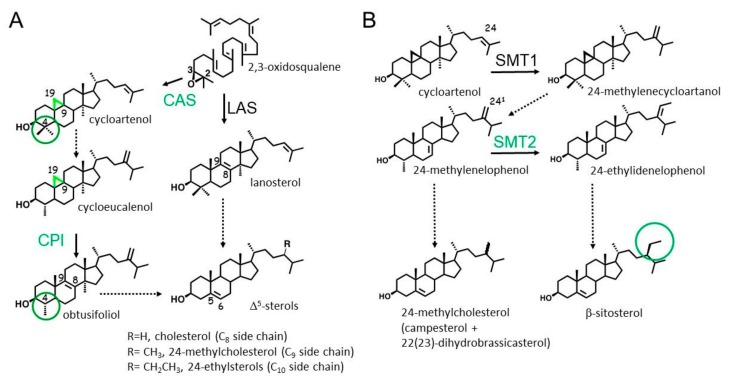
A simplified scheme of phytosterol biosynthesis pointing out major peculiarities of the pathway. (**A**) 2,3-Oxidosqualene cyclization into 9β,19-cyclopropylsterols (cycloartenol further converted into cycloeucalenol), then into obtusifoliol, and finally into -^5^-sterols. Green circles highlight 4,4-dimethylsterols and 4-methylsterols in plants [18], other plant-specific features appear in green in this scheme; (**B**) non-consecutive side chain methylation reactions of cycloartenol by SMT1 and of 24-methylenelophenol by SMT2, leading to 24-methylcholesterol and β-sitosterol. The ratio of epimeric 24-methylcholesterol molecules campesterol/ 22(23)-dihydrobrassicasterol is close to 6:4 in higher plants [19,20]. CAS, cycloartenol synthase; LAS, lanosterol synthase; CPI, cyclopropyl isomerase; SMT1, S-adenosyl-L-Met-cycloartenol-C24-methyltransferase; SMT2, S-adenosyl-L-Met-24^1^-methylenelophenol-C24-methyltransferases. Common sterol nomenclature of sterols is used. An accurate sterol nomenclature can be found in Moss [21] and Nes [3]. Each arrow is an enzymatic step. Dashed arrows represent more than one enzymatic step.

**Figure 2 molecules-25-01111-f002:**
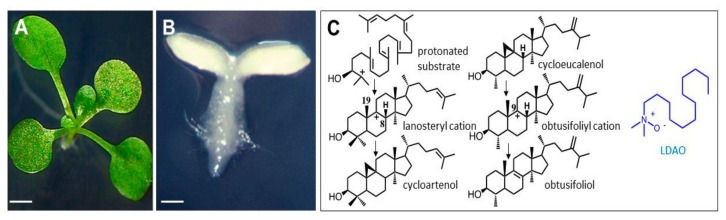
*Arabidopsis**thaliana* seedlings grown on synthetic medium containing LDAO. (**A**) two-week-old control seedling, scale bar = 2 mm; (**B**) two-week-old seedling after seed germination in the presence of 20 mg.L^−1^ LDAO, scale bar = 0.5 mm; (**C**) left, reaction pathway for the CAS-mediated cyclization of the C-2 protonated form of 2,3-oxidosqualene into a lanosteryl cation stabilized into cycloartenol, after proton abstraction from C-19 (note that LAS-mediated proton abstraction at C-8 yields lanosterol); middle, reaction pathway for the CPI-mediated isomerization of cycloeucalenol as a postulated C9-carbonium ion (obtusifoliyl cation) stabilized into obtusifoliol; right, structure of LDAO, a zwitterionic amine N-oxide.

**Figure 3 molecules-25-01111-f003:**
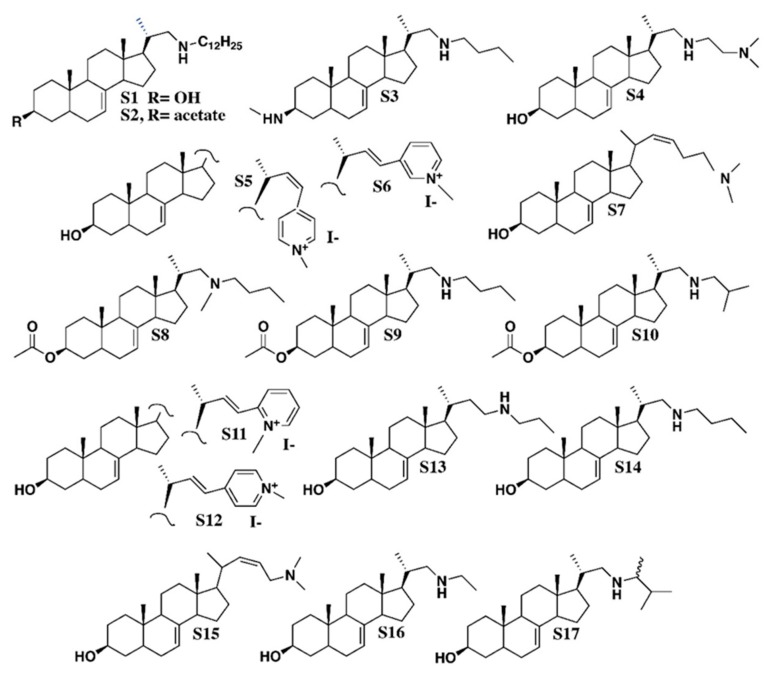
Chemical structures of azasteroids. The synthesis of all compounds is described in Renard et al. [40]. Nomenclature of S1 to S17 is given in the methods section, according to Renard et al. [40].

**Figure 4 molecules-25-01111-f004:**
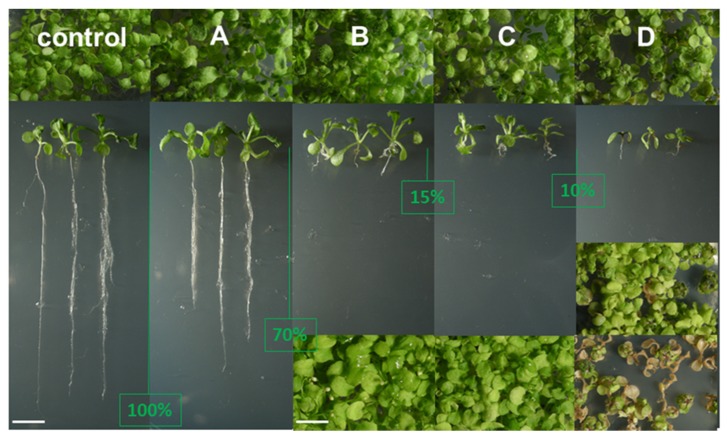
*Arabidopsis thaliana* morphogenetic inhibition by azasteroid treatment. Seedlings were photographed as top views of cultures or as whole seedlings vertically aligned for clear visualization of root lengths after careful extraction of the root system from the synthetic medium. (**A**–**C**), and (**D**) show different strengths of compounds grouped as described in the text. The topline of images shows the aspect of two-week-old cultures. Whole seedlings vertically aligned are from the same two-week-old cultures. The reduction in root length is indicated by the green vertical line and the value in percentage of the control root length. The bottom line of images for (**B**–**D**) type of growth effect show cultures photographed four weeks after the onset of germination. In the case of D, the top image shows the effect of compound S14 and the toxic effect of compound S13. Growth phenotypes categories correspond to sterol profiles (**A**–**D**) highlighted in Figure 5 and further detailed in Table 2.

**Figure 5 molecules-25-01111-f005:**
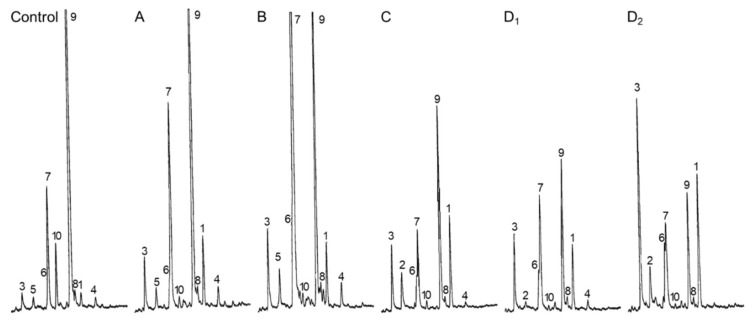
Sterol profiles of whole seedlings treated with side chain azasteroids. GC-FID traces of representative profiles for each category of compounds shown in Figure 3. (**A**) compounds S5 to S10; (**B**), compounds S15, S16, S17; (**C**), compounds S11 and S12; (**D_1_**), compound S14; and (**D_2_**), compound S13. Peaks are labeled as in Table 2: 1, cycloartenol; 2, desmosterol; 3, cholesterol; 4, 24-methylenecycloartanol; 5, brassicasterol; 6, 24-methylenecholesterol; 7, 24-methylcholesterol (campesterol and its epimer 22(23)-dihydrobrassicasterol; 8, isofucosterol; 9, sitosterol; 10, stigmasterol; minor compounds are not annotated for clarity of the figure.

**Figure 6 molecules-25-01111-f006:**
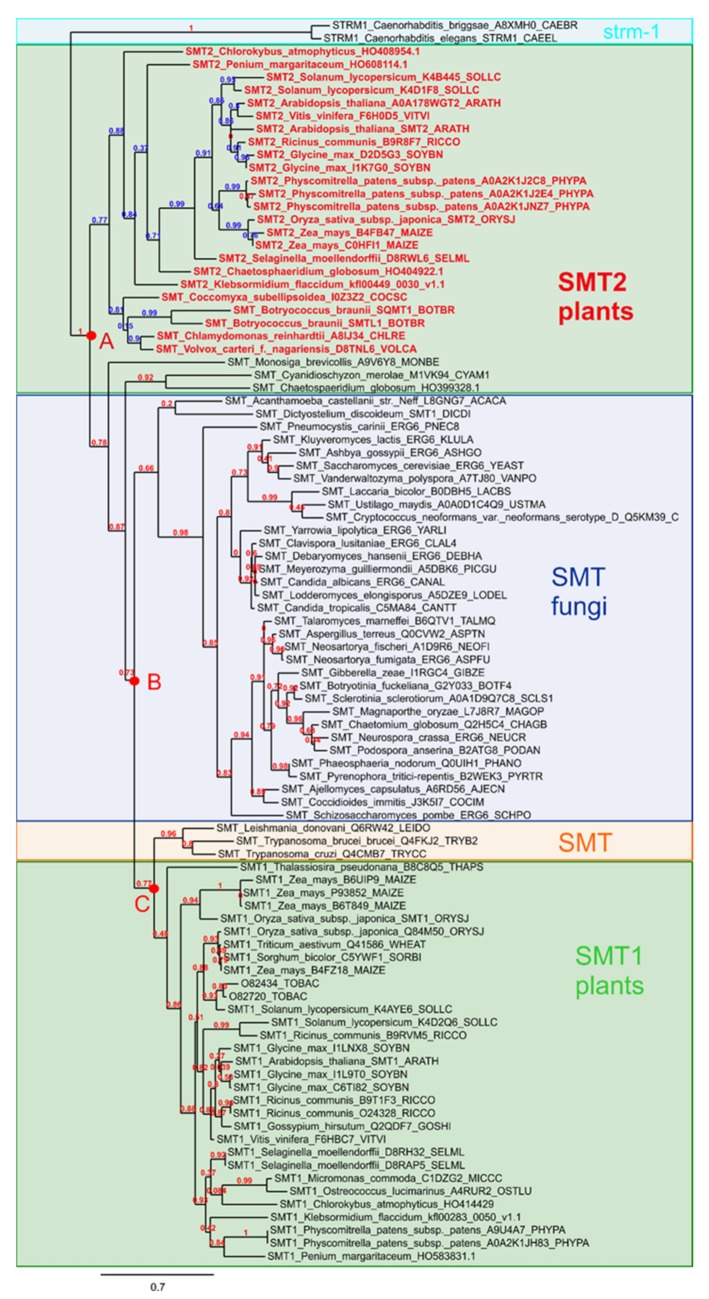
Phylogenetic tree of SMTs. Sequences were retrieved from the Uniprot database using sequences previously characterized. The sequences were aligned using MAFFT and PHYML for the phylogenetic inference.

**Figure 7 molecules-25-01111-f007:**
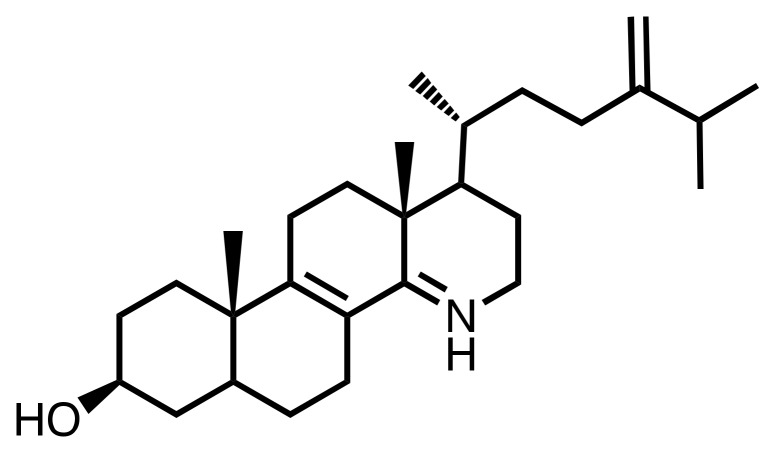
Structure of the natural 15-azasterol from *Geotrichum flavo-brunneum*.

**Table 1 molecules-25-01111-t001:** Sterol composition of LDAO (C_14_H_31_NO, lauryldimethylamine oxide, and N,N-dimethyldodecylamine N-oxide) treated seedlings.

LDAO mg·L^−1^	0	10	20
2,3-Oxidosqualene in mg·g^−1^ dry weight	tr	0.2	1.4
Sterols%			
Cycloartenol	1.0	Nd	Nd
24-Methylenecycloartanol	2.0	2.0	5.0
Cycloeucalenol	0.3	48.0	51.0
Cyclofontumienol	nd	3.0	2.0
24-Methylenepollinastanol	nd	27.0	25.2
24-Methylpollinastanol	nd	1.0	Nd
**Total 9β,19-cyclopropylsterols**	**3.3**	**81.0**	**83.2**
Cholesterol	4.5	0.3	Nd
24-Methylenecholesterol	1.2	Nd	Nd
Campesterol	13.5	1.0	0.8
Isofucosterol	2.3	1.0	1.2
Sitosterol	69.5	9.5	9.5
Stigmasterol	3.0	1.0	1.3
**Total 24-alkyl-5-sterols**	**94.0**	**12.8**	**12.8**
Other minor sterols	2.7	6.2	4.0
Total sterols in mg·g^−1^ dry weight	2.2	1.2	1.7

Note: tr, trace amounts; nd, not detected. Analyses of two biological replicates produced very similar results.

**Table 2 molecules-25-01111-t002:** Sterol composition of *Arabidopsis* seedlings grown on various azasterols classified as groups A, B, C, and D. Representative profiles are shown for one or two compounds of each group.

			Group A		Group B		Group C	Group D	
Sterols %		control	S6	S8	S16	S17	S11	S13	S14
Cycloartenol (1)		1.0	14.0	5.1	4.5	3.0	16.3	20.3	15.0
Desmosterol (2)		nd	1.8	Nd	Nd	nd	7.5	5.3	1.0
Cholesterol (3)		2.0	8.5	4.0	5.3	1.8	10.5	28.7	14.5
**Total C_8_ side chain sterols**		**3.0**	**24.3**	**9.1**	**9.8**	**4.8**	**34.3**	**54.3**	**30.5**
24-Methylenecycloartanol (4)		0.6	0.5	1.4	1.3	1.2	nd	Nd	1.0
Brassicasterol (5)		1.1	nd	2.3	2.3	2.4	nd	0.9	Nd
24-Methylenecholesterol (6)		nd	3.7	*	*	*	7.0	8.7	7.5
24-Methylcholesterol (7)		14.5	21.2	27.1	59.0	42.4	15.0	16.0	21.5
**Total C_9_ side chain sterols**		**16.2**	**25.4**	**30.8**	**62.6**	**46.0**	**22**	**25.6**	**30.0**
Isofucosterol (8)		0.9	1.4	0.8	0.6	0.5	1.5	0.9	0.5
Sitosterol (9)		73.9	48.0	58.1	26.5	47.8	40.8	18.2	38.5
Stigmasterol (10)		6.0	0.9	1.2	0.5	0.9	1.4	1.0	0.5
**Total C_10_ side chain sterols**		**80.8**	**50.3**	**60.1**	**27.6**	**49.2**	**43.7**	**20.1**	**39.5**

nd, not detected; *, low amount integrated into the 24-methylcholesterol peak (7); GC traces shown in Figure 5 are sterol profiles from control, S8 (Group A), S16 (Group B), S11 (Group C), S14 (Group D), and S13 (Group D) treated plants. Analyses of two biological replicates produced very similar results.

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
