# Peer review of "Inhibition of Phytosterol Biosynthesis by Azasterols"

_molecules, 2020, doi:10.3390/molecules25051111_

Round 1

Reviewer 1 Report

The manuscript introduces into the phytochemical analysis of sterols from Arabidopsis thaliana that are reacted with a series of azasterols, revealing specific in vivo inhibition of sterol methyl transferasas, by acting as branch point between the campesterol and sitosterol biosynthetic segments in the pathway. Side chain azasteroids that modify the sitosterol homeostasis may assist the enhancing of its particular function in the plant development. The document is well written, the experimental is solid and discussion and conclusions are consistent to the results obtained. Minor corrections are suggested, as follows:

Correct the multiple citations to “Renard et al. [35]. “ (i.e., avoid comma after the last name and before et al.)

In figure 2 use full name of studied plant (i.e., “Arabidopsis thaliana”). Keep in mind to make figures and tables as more self-explainable as possible.

Figure 3 should have its caprtion at the bottom (revise format of document)

In figure 4 caption “The reduction in root length is indicated by the green vertical line and the value in percentage of the control root length.”

Author Response

Academic Editor Comments

I think the content of this manuscript is appropriate for the special issue on "Natural Sterols". However, confirm the structural formula of the natural 15-azasterol shown in Figure 7. I think it is necessary to have the angular methyl groups (C-18 and C-19).

We thank the academic editor for his appreciation of the present work. The mistake in the formula of 15-azasterol is corrected, we apologize for such a mistake.

Reviewer 1

The manuscript introduces into the phytochemical analysis of sterols from Arabidopsis thaliana that are reacted with a series of azasterols, revealing specific in vivo inhibition of sterol methyl transferasas, by acting as branch point between the campesterol and sitosterol biosynthetic segments in the pathway. Side chain azasteroids that modify the sitosterol homeostasis may assist the enhancing of its particular function in the plant development. The document is well written, the experimental is solid and discussion and conclusions are consistent to the results obtained. Minor corrections are suggested, as follows:

- We thank Reviewer 1 for his/her favorable evaluation of our work.

Correct the multiple citations to "Renard et al. [35]. "(i.e., avoid comma after the last name and before et al.)

- The citation of the article Renard et al. [35] has been replaced in the text by the number only [35], or was modified as suggested (the comma after the last name was deleted).

In figure 2 use full name of studied plant (i.e., "Arabidopsis thaliana"). Keep in mind to make figures and tables as more self-explainable as possible.

- In the legend for Figure 2, A.thaliana is replaced by Arabidopsis thaliana. In addition, the size of scale bars in A and B panels of Figure 2 are indicated as follows: .. (A) .. , scale bar = 2 mm; (B) .., scale bar = 0.5 mm. We thank Reviewer 1 for allowing us to make this clarification.

Figure 3 should have its caprtion at the bottom (revise format of document)

- The legend for Figure 3 is now placed at the bottom of the figure.

In figure 4 caption "The reduction in root length is indicated by the green vertical line and the value in percentage of the control root length."

- The text for Figure 4 legend has been modified: Arabidopsis thaliana (genus and species), and the word 'percentage' replaces the sign %.

Once again, we thank Reviewer 1 for sharing comments, and we do hope his/her concerns are addressed. 

Reviewer 2 Report

The manuscript by Darnet and co-workers addresses the inhibition of phytosterol biosynthesis in Arabidopsis thaliana by lauryldimethylamine oxide (LDAO) and a series of azasteroids. These two classes of compounds were previously reported as inhibitors of steroid biosynthesis. LDAO is an inhibitor of 2,3-epoxysqualene-sterol cyclases. The azasteroids with N atom in the aliphatic side chain were reported to be inhibitory to sterol C24 methyltransferase in fungi. The authors now focused on the inhibitory effects of these compounds in Arabidopsis thaliana. Through the sterol profiles of compound treated plant samples by GC-FID and GC-MS analysis, the authors observed there were certain steroids accumulated in the presence of testing compounds and concluded that LDAO was inhibiting cycloartenol synthase and cyclopropyl isomerase in the plants and three azasteroids were inhibiting the SMT2 in the plants specifically while the remaining azasteroids tested had multiple targets in the plants. The authors claimed that the identified azasteroids targeting SMT2 in the plant provided a class of powerful tool compounds to control the production of sitoserol and to characterize sitosterol’s effect on root development given the phenotype that was observed in this study. 

The data presented appear solid. However, there are no standard deviations presented for all of the quantification analyses in Table 1 and Table 2. The major conclusion drawn from the results is the specific inhibition of SMT2 by compounds 15-17 in Arabidopsis thaliana. This is supported by the similar sterol profile observed from the SMT2 loss of function mutant (Ref 15,17). However, the other conclusions that were drawn only based on the sterol profile seem to be weak. The authors claimed that compound group C showed inhibitory effects on both SMT1 and SMT2, and group D inhibited SMT1. These claims are only based on the analysis of accumulated steroid. In fact, the authors claimed that plant treated with compound D had a sterol profile depleted in 24-methylsterols. This is not correct. As shown in Table2, group D showed similar levels of 24-methylsterols to group A and C at ~20-30%. They are not confirmed either in enzyme assays or gene knock out mutants. It is possible that these observations could be caused from some down-stream effect of another enzyme inhibition in the pathway. In addition, the novelty of the study is rather critical, as previous work has suggested that the same type of compounds are inhibitors of the similar enzymes in fungi.

Some minor points:

The numbering of subtitles is not correct.  The abbreviation for Arabidopsis thaliana should be consistent throughout.  Figure 3 was embedded in the middle of a paragraph and the legend should be under the figure. Page 7, from “the in vivo efficient” to the end of this paragraph, this should go to the discussion.  Page 12, “…but currently characterized” is this currently being characterized? Or have been characterized. Please rephrase. 

Author Response

Academic Editor Comments

I think the content of this manuscript is appropriate for the special issue on "Natural Sterols". However, confirm the structural formula of the natural 15-azasterol shown in Figure 7. I think it is necessary to have the angular methyl groups (C-18 and C-19).

We thank the academic editor for his appreciation of the present work. The mistake in the formula of 15-azasterol is corrected, we apologize for such a mistake.

Reviewer 2

Open Review

English language and style

( ) Extensive editing of English language and style required
( ) Moderate English changes required
(x) English language and style are fine/minor spell check required
( ) I don't feel qualified to judge about the English language and style

Yes

Can be improved

Must be improved

Not applicable

Does the introduction provide sufficient background and include all relevant references?

(x)

( )

( )

( )

Is the research design appropriate?

(x)

( )

( )

( )

Are the methods adequately described?

( )

(x)

( )

( )

Are the results clearly presented?

(x)

( )

( )

( )

Are the conclusions supported by the results?

( )

(x)

( )

( )

The manuscript by Darnet and co-workers addresses the inhibition of phytosterol biosynthesis in Arabidopsis thaliana by lauryldimethylamine oxide (LDAO) and a series of azasteroids. These two classes of compounds were previously reported as inhibitors of steroid biosynthesis. LDAO is an inhibitor of 2,3-epoxysqualene-sterol cyclases. The azasteroids with N atom in the aliphatic side chain were reported to be inhibitory to sterol C24 methyltransferase in fungi. The authors now focused on the inhibitory effects of these compounds in Arabidopsis thaliana. Through the sterol profiles of compound treated plant samples by GC-FID and GC-MS analysis, the authors observed there were certain steroids accumulated in the presence of testing compounds and concluded that LDAO was inhibiting cycloartenol synthase and cyclopropyl isomerase in the plants and three azasteroids were inhibiting the SMT2 in the plants specifically while the remaining azasteroids tested had multiple targets in the plants. The authors claimed that the identified azasteroids targeting SMT2 in the plant provided a class of powerful tool compounds to control the production of sitoserol and to characterize sitosterol's effect on root development given the phenotype that was observed in this study. 

- We thank Reviewer 2 for his/her positive assessment of our work.

The data presented appear solid. However, there are no standard deviations presented for all of the quantification analyses in Table 1 and Table 2.

- Reviewer 2 is right, and there are no standard deviations presented in analyses shown in Table 1 and Table 2. In the Material and Methods section, in 1.2 Plant treatment with chemicals, we have indicated that: "Experiments were done twice independently and in a triplicate assay for each compound. One of these triplicates in each biological assay was analyzed to determine its sterol profile." This means that the morphological effect of the chemicals on seedlings were assessed in two distinct experiments (3 Petri plates for a given compound), and for these experiments, 1 of the 3 Petri plates was sampled for sterol analysis. Consequently, sterol analysis were performed twice for one given treatment. We have added the sentence "Analysis was done for two biological replicates and produced very similar results" at the bottom of Table 1 and Table 2.

The major conclusion drawn from the results is the specific inhibition of SMT2 by compounds 15-17 in Arabidopsis thaliana. This is supported by the similar sterol profile observed from the SMT2 loss of function mutant (Ref 15,17).

- We agree with Reviewer 2 on this conclusion. The fact that seedlings accumulate higher amounts of campesterol at the expense of sitosterol indicates a strong inhibition of SMT2. Because this SMT2 enzyme is downstream of the SMT1 enzyme in the sterol pathway, it also indicates the inefficiency of compounds S15-S17 to block SMT1.

However, the other conclusions that were drawn only based on the sterol profile seem to be weak. The authors claimed that compound group C showed inhibitory effects on both SMT1 and SMT2, and group D inhibited SMT1. These claims are only based on the analysis of accumulated steroid. In fact, the authors claimed that plant treated with compound D had a sterol profile depleted in 24-methylsterols. This is not correct. As shown in Table2, group D showed similar levels of 24-methylsterols to group A and C at ~20-30%. They are not confirmed either in enzyme assays or gene knock out mutants. It is possible that these observations could be caused from some down-stream effect of another enzyme inhibition in the pathway. In addition, the novelty of the study is rather critical, as previous work has suggested that the same type of compounds are inhibitors of the similar enzymes in fungi.

- We thank Reviewer 2 for pointing out this mistake. The text is revised as follows:

"Seedlings from the groups C and D exhibited a rather moderate increase in the amount of 24-methylcholesterol,  a decrease in sitosterol and an increase in cycloartenol and cholesterol, a chemical profile indicative of a possible dual inhibitory effect on both SMT1 and SMT2 (Table 2, Figure 5C). Furthermore, seedlings from category D (S13 and S14) had a very strong enrichment in cycloartenol and cholesterol, the major C8 side chain sterols (Table 2, Figure 5D). The predominance of cycloartenol and cholesterol in such sterol profiles, which is reminiscent of the sterol profile of a loss-of-function allele of SMT1[Diener et al. 2000;44], provides conclusive evidence of strong inhibition of SMT1 by group D compounds. In such smt1-1 Arabidopsis thaliana mutant, the proportion of C9-side chain sterols is slightly higher in the mutant (31%) versus the wild-type (25%) [Diener et al. 2000; 44]. Consequently, the increase in 24-methyl(ene)sterols in groups C and D could also be caused by yet unspecified effects downstream to SMT1."

Diener et al. 2000 is a new reference numbered as [44].

- We agree with Reviewer 2 that an additional series of experiments will be necessary for such a project to ultimately characterize specific interaction of some azasteroids with SMT1 vs. SMT2. This will require the measurements of enzyme activities in subcellular fractions from plants (microsomes) or yeast expressing SMT1 and SMT2 from plants, this will require the purification of enzymes for further binding assays, and so on, in the presence of various ligands. At this stage of the project, these experiments are in front of us. It is our purpose in this short paper to point out the very specific effect of some of the azasteroids on SMT2 only. We also agree on the fact that previous work from different groups had illustrated the inhibition of SMT1 and ERG6 (yeast) by azasteroids, including those used in the present study, but not on plants (this particular point is mentioned page 6 lines 2-3 (…Sterol analogs S1 to S17 (Figure 3), already known as inhibitors of fungal sterol-C24-methyltransferase  [35], were fed…).

Some minor points:

The numbering of subtitles is not correct. 

- The numbering of subtitles is corrected.

The abbreviation for Arabidopsis thaliana should be consistent throughout. 

- Arabidopsis thaliana is used throughout.

Figure 3 was embedded in the middle of a paragraph and the legend should be under the figure.

- Legend for Figure 3 is under the figure.

Page 7, from "the in vivo efficient" to the end of this paragraph, this should go to the discussion. 

- We thank Reviewer 2 for this suggestion. We would like however, to keep this part of 'Results' (the sentences refer to Figure 4D) as it is since it is a good transition towards a discussion paragraph. Thank you very much for your understanding.

Page 12, "…but currently characterized" is this currently being characterized? Or have been characterized. Please rephrase. 

- We have rephrased the sentence, as follows: In fact, chemicals that can be exploited in chemotherapy, especially to fight pathogenic fungi like Cryptococcus, but also protozoa like Trypanosoma or Leishmania, have been characterized [25, 50-52].

- Additional typos corrected by the authors in this review process:

Page 2, first line, 24-methylsterols is changed to 24-methyl(ene)sterols

Legend for Figure 1: 4th line from botton, 24 and 241 in the nomenclature of SMTs enzymes are properly placed.

Legend for Figure 5: nomenclature of peaks appearing in the GC trace was not complete: … 9, sitosterol; 10, stigmasterol ..

- Once again we thank Reviewer 2 for his/her valuable comments.

Reviewer 3 Report

Darnet and colleagues reported new inhibitors of SMT2, an important sterol biosynthetic enzyme in plants. Inhibitor treatment of Arabidopsis revealed interesting phenotypes. In my opinion, the findings from this study are informative and will advance our knowledge on enzymology. Some minor comments are listed as follows:

1) In Introduction, “fungi sterols” should be “fungal sterols”.

2) In the last paragraph of Introduction, “… are reported’ should be “… is reported”. Perhaps, the last sentence “The specific in vivo inhibition … distinct sterol profiles in plants” can be improved by rephrasing.

3) Was any internal standard used in sterol analysis? If so, please state.

4) In Section 1.5, remove “Renard, et al.” after “sterol C24-methyltransferase”.

5) In Section 1.5, “dramatic hampered rosette development” should be “dramatically hampered rosette development”.

6) In Discussion, remove “are” in the statement “these enzymes are maybe phylogenetically …”.

7) In Figure 6 legend, is the long list of sequence names and accession numbers necessary, given the duplication as they are stated in the phylogenetic tree?

Author Response

Academic Editor Comments

I think the content of this manuscript is appropriate for the special issue on "Natural Sterols". However, confirm the structural formula of the natural 15-azasterol shown in Figure 7. I think it is necessary to have the angular methyl groups (C-18 and C-19).

We thank the academic editor for his appreciation of the present work. The mistake in the formula of 15-azasterol is corrected, we apologize for such a mistake.

Reviewer 3

  • In Introduction, "fungi sterols" should be "fungal sterols".

"Fungal" is replacing "fungi".

  • In the last paragraph of Introduction, "… are reported' should be "… is reported". Perhaps, the last sentence "The specific in vivo inhibition … distinct sterol profiles in plants" can be improved by rephrasing.

"Are reported" is replaced by "is reported".

The last sentence "The specific in vivo inhibition of the land plant enzyme SMT2 by some azasterols is setting these latter as valuable probes to study at multi-scale levels (organ, tissue, cell-type, cell compartments) the function of distinct sterol profiles in plants" is rephrased as follows:

The specific in vivo inhibition of the land plant enzyme SMT2 by some azasterols is setting these latter as valuable probes to study the function of distinct phytosterol profiles at multi-scale levels.

  • Was any internal standard used in sterol analysis? If so, please state.

Quantification of steryl acetates in mg was based on an internal standard (betulyl diacetate) as described in ref 37. Therefore, the sentence is the Materials and Methods section 1.3. Sterol analysis "Steryl acetate detection and analysis were performed as described [36, 37]." is changed to Steryl acetate detection, analysis, and quantification were performed as described [36, 37]. Reference [37] is Babyichuk et al 2008 Proc Natl Acad Sci USA, not Babyichuk et al 2008 Plant Signal. Behaviour, this mistake has been corrected.

  • In Section 1.5, remove "Renard, et al." after "sterol C24-methyltransferase".

Page 6 line 3 the change is done.

  • In Section 1.5, "dramatic hampered rosette development" should be "dramatically hampered rosette development".

This mistake is corrected, we thank Reviewer for helping us to improve the manuscript.

  • In Discussion, remove "are" in the statement "these enzymes are maybe phylogenetically …".

"Are" is deleted. We thank the Reviewer for helping us to improve the manuscript.

  • In Figure 6 legend, is the long list of sequence names and accession numbers necessary, given the duplication as they are stated in the phylogenetic tree.

Reviewer 3 is right; the long list of sequence names and accession numbers is not absolutely necessary in the legend for Figure 6. Therefore it is deleted.

We thank Reviewer 3 for his/her contribution to the overall improvement of the paper.